# Lacosamide adjunctive therapy for partial-onset seizures: a meta-analysis

Sonja C. Sawh[1], Jennifer J. Newman[2], Santosh Deshpande[1] and Philip M. Jones[3,4]

[1] Evidence-Based Medicine/Drug & Therapeutics Committee Resource, London Health Sciences Centre, Pharmacy Department, University Hospital, London, ON, Canada
[2] Evidence-Based Medicine, London Health Sciences Centre, Victoria Hospital, London, ON, Canada
[3] Department of Anesthesia & Perioperative Medicine, University of Western Ontario, London, ON, Canada
[4] Department of Epidemiology & Biostatistics, University of Western Ontario, London, ON, Canada

Corresponding author
Sonja C. Sawh,
sonja.sawh@lhsc.on.ca

## ABSTRACT

**Background.** The relative efficacy and safety of lacosamide as adjunctive therapy compared to other antiepileptic drugs has not been well established.

**Objective.** To determine if lacosamide provides improved efficacy and safety, reduced length of hospital stay and improved quality of life compared with other anti-epileptic therapies for adults with partial-onset seizures.

**Data Sources.** A systematic review of the medical literature using Medline (1946–Week 4, 2012), EMBASE (1980–Week 3, 2012), Cochrane Central Register of Controlled Trials (Issue 1 of 12, January 2012). Additional studies were identified (through to February 7, 2012) by searching bibliographies, the FDA drug approval files, clinical trial registries and major national and international neurology meeting abstracts. No restrictions on publication status or language were applied.

**Study Selection.** Randomized controlled trials of lacosamide in adults with partial-onset seizures were included.

**Data Extraction.** Study selection, extraction and risk of bias assessment were performed independently by two authors. Authors of studies were contacted for missing data.

**Data Synthesis.** All pooled analyses used the random effects model.

**Results.** Three trials (1311 patients) met inclusion criteria. Lacosamide increased the 50% responder rate compared to placebo (RR 1.68 [95% CI 1.36 to 2.08]; $I^2 = 0\%$). Discontinuation due to adverse events was statistically significantly higher in the lacosamide arm (RR3.13 [95% CI 1.94 to 5.06]; $I^2 = 0\%$). Individual adverse events (ataxia, dizziness, fatigue, and nausea) were also significantly higher in the lacosamide group.

**Limitations.** All dosage arms from the included studies were pooled to make a single pair-wise comparison to placebo. Selective reporting of outcomes was found in all of the included RCTs.

**Conclusions.** Lacosamide as adjunctive therapy in patients with partial-onset seizures increases the 50% responder rate but with significantly more adverse events compared to the placebo.

## INTRODUCTION

Epilepsy affects 15,500 new Canadians annually (*Epilepsy Canada, 2011*) with partial-onset seizures being the most common seizure type in adults - affecting up to 60% of adults who have epilepsy (*Epilepsy Canada, 2011*). Up to one-third of newly-diagnosed patients are refractory to drug therapy and this presents a therapeutic challenge (*Beyenburg, Stavem & Schmidt, 2010*). Adjunctive therapy with antiepileptic drugs (AEDs) is the standard of care for patients with refractory epilepsy (*French, Kanner & Bautista, 2004*). However, current guidelines (*French, Kanner & Bautista, 2004*) do not address the more recently-available AEDs, including lacosamide, for the treatment of refractory epilepsy.

Lacosamide is a novel AED, consisting of a functionalized amino acid molecule believed to stabilize hyperexcitable neuronal membranes and inhibit repetitive neuronal firing (*Lexi-Drugs, 2011*). Health Canada has approved lacosamide for use as adjunctive therapy in the management of partial-onset seizures in adult patients with epilepsy who are not satisfactorily controlled with conventional therapy (*Canadian Pharmacists Association, 2011*).

All previously-published systematic reviews of lacosamide (*Beyenburg, Stavem & Schmidt, 2010*; *Beydoun et al., 2009*; *Chung et al., 2010a*; *Simoens, 2011*; *Costa et al., 2011*; *Ryvlin, Cucherat & Rheims, 2011*) have concluded that lacosamide is efficacious in reducing seizure frequency compared to placebo, but each review had methodological challenges limiting its interpretability. To better estimate the effect size of lacosamide, this systematic review was designed to include all doses of lacosamide studied, using the intention to treat population, and considering all important outcomes, in addition to closely examining lacosamide's adverse events (which have not been adequately explored in the previous reviews).

The objective of this systematic review was to determine the relative benefits and harm of lacosamide therapy compared to other AEDs or placebo, as adjunctive therapy for adults with partial-onset seizures.

## METHODS

### Protocol and registration

The search strategy, methods of analysis and inclusion criteria were specified in advance and documented in a protocol. The protocol for this systematic review was registered with the Prospective International Register of Systematic Reviews (PROSPERO) and can be found online (*Sawh & Newman, 2012*).

## Information sources/search strategy

Studies were identified by searching the following electronic databases: Medline (OVID 1946 to Week 4, 2012), EMBASE (OVID, 1980 to Week 3 2012), Cochrane Central Register of Controlled Trials (CENTRAL) (Wiley Issue 1 of 12, January 2012).

We contacted the manufacturer of lacosamide and experts in the field for information about unpublished or ongoing studies. The Food and Drug Administration's (FDA) Approved Drug Products database was searched for clinical trials used to support marketing approval and/or labelling changes in the United States. Conference abstracts and posters were searched from selected meetings of the American Epilepsy Society, World Congress of Neurology, International Epilepsy Congress, and the European Congress on Epileptology. We also searched the metaRegister of Controlled Trials (*m*RCT) to identify ongoing trials.

Reference lists of all retrieved studies were reviewed for additional relevant studies.

The search was developed and conducted by one of the authors (SS) and reviewed by a Research Librarian (KC). The last search was run February 7, 2012. We used the following search terms to search all trial registers and databases (modified to suit each specific database): randomized controlled trials, epilepsy, seizures, partial epilepsy, lacosamide, and Vimpat. No language restrictions were imposed on the electronic database searches. The online protocol provides the detailed search strategy used in this review.

## Study selection

Title and abstract screening was conducted in duplicate to identify potentially eligible papers using a standardized guide for trial inclusion based on title and abstract screening. Two reviewers (JN and SS) underwent a calibration process to identify potential discrepancies in interpretation of the form (with the first 100 citations as a sample). Publications that could possibly have met the selection criteria were retrieved as full-text articles and examined.

Full-text screening was conducted, independently by two reviewers, to confirm eligibility using a standardized screening form (Table S1). We used Fleiss and Cohen's weighted Kappa (using the program Kappa.exe (*Cyr & Francis, 1992*)) to assess agreement between the two reviewers on the selection of full-text articles for inclusion (*Fleiss & Cohen, 1973*). All disagreements were resolved by discussion.

We documented the study selection process in a flow chart as recommended in the PRISMA statement (*Liberati et al., 2009*) showing the total numbers of retrieved references and the numbers of included and excluded studies, and the reasons for exclusion (Fig. 1).

## Data collection process & data items

Data were extracted independently by two reviewers (JN and SS) using an *a priori* standardized data extraction form with the aid of a data and validity extraction manual. The two sets of extracted data were compared and all discrepancies were resolved by discussion. Data was extracted from each included trial on the following general areas of information:

**Peer**J _________

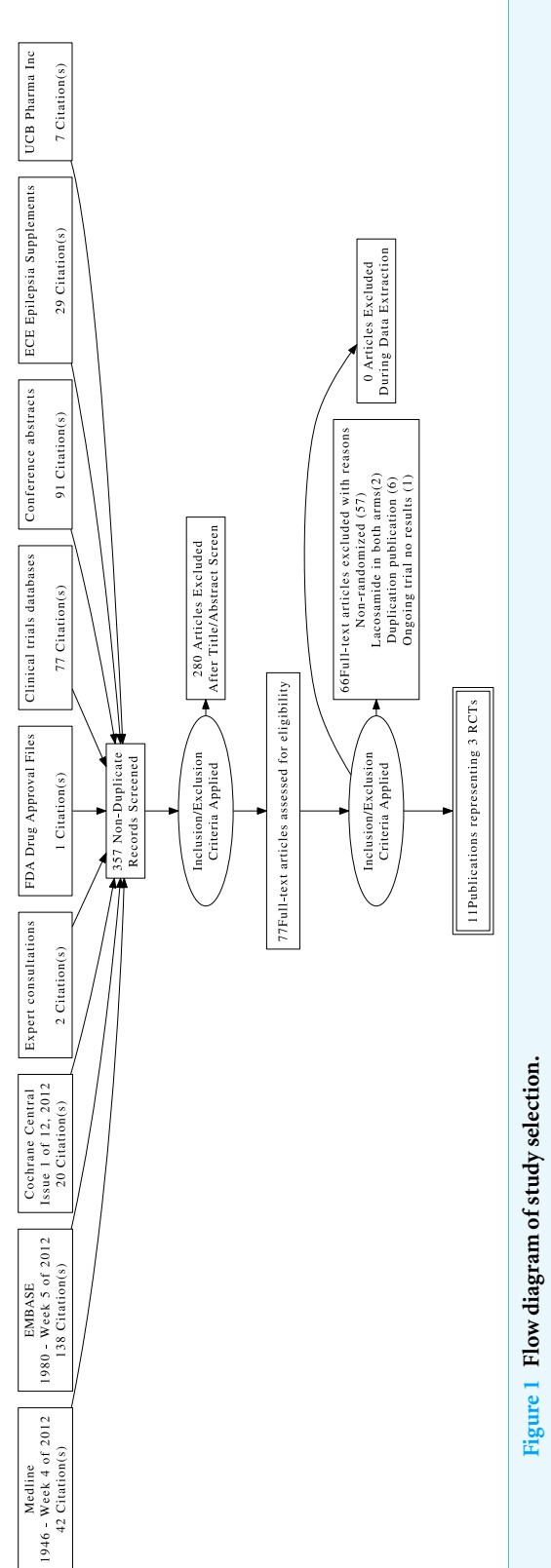

**Figure 1 Flow diagram of study selection.**

### Trial characteristics
- Number of participating centres and countries
- Inclusion criteria
- Exclusion criteria
- Number of patients eligible and randomized
- Treatment duration and length of follow-up of patient outcomes
- Data collection time points
- Treatment arms in the trials
- Ethics review board approval and patient consent to participate
- Funding source

### Participant characteristics
- Number of patients randomized and with available outcome data
- Epilepsy diagnosis
- AED use (number and types)

### Primary and secondary outcomes
- Outcome definition
- Direction of outcome (i.e., harm or benefit)
- Time point(s) of outcome evaluation
- Outcome unit of measurement and measure of error (if continuous). Where possible, for continuous measures, mean outcome values and standard deviations were recorded or determined as measurements of outcome.

Study authors were contacted by e-mail to request information about missing data for included trials. For studies with multiple publications, all versions of the study were reviewed to ensure complete access to maximal trial data. In the event of inconsistency of study data between multiple publications (for example, between a Food and Drug Administration submission and a peer-reviewed paper published in a journal), the peer-reviewed publication was used as the primary data set.

## Risk of bias in individual studies
Two reviewers (JN and SS) independently assessed the risk of bias for each included study using the criteria outlined in the *Cochrane Handbook for Systematic Reviews of Interventions* (*Higgins, Altman & Sterne, 2011*). Reviewers were not blinded to the study authors, journal or outcome data. We specifically assessed the trial characteristics as specified in the protocol.

- sequence generation;
- allocation concealment;
- blinding of the study (participants, personnel, outcome assessors, data collectors, data analysts) as defined by *Akl et al. (2012)*;

- incomplete outcome data;
- selective outcome reporting;
- other sources of bias.

A summary table and a graph for risk of bias were created using Review Manager software 5.1 (*The Nordic Cochrane Centre, 2011*).

## Synthesis of results

We calculated the pooled relative risks (RRs) and 95% confidence intervals (CI) for dichotomous variables using the Mantel-Haenszel method (*Deeks, Higgins & Altman, 2011*). For continuous variables measured using the same scales, the mean differences (MD) and its 95% CI were calculated using the inverse variance method. If a continuous outcome variable was measured using different scales across studies, we calculated the standardized mean difference (SMD).

All of our analyses included the total numbers of participants in the treatment groups to which they had been allocated (intention to treat analysis). Participants not completing follow up or with inadequate seizure data were assumed to be non-responders.

We contacted study authors for clarification if more information was needed, and to request missing data.

Randomized trials included multiple dosages of lacosamide in separate randomized arms. For the purpose of the meta-analysis, all lacosamide dosages were combined into one "lacosamide" arm (*Cochrane Handbook for Systematic Reviews of Interventions, 2011*).

We tested statistically for heterogeneity with a chi-square test and used $I^2$ to measure inconsistency (the percentage of total variation across studies due to heterogeneity). We used "small," ($\leq 25\%$), "moderate" (between 25% and 50%) and "large" ($\geq 50\%$) to describe the statistical heterogeneity as measured by $I^2$ (*Higgins et al., 2003*). Forest plots were visually inspected for possible sources of heterogeneity.

A summary of findings table was created using GRADEpro software for the three primary outcomes of this review (*Brozek, Oxman & Schünemann, 2008*). We planned to assess the possibility of publication bias by using funnel plots (*Egger & Davey Smith, 1995*).

## Additional analyses

The following subgroup analyses were pre-specified for primary outcomes: patients younger than 18 years old (if the pediatric outcome data was reported as a discrete subgroup), placebo vs. active comparators, intravenous vs. oral lacosamide, and comparing studies with high vs. low risk of bias. Post-hoc, the potential of a dose-response relationship of lacosamide was explored using subgroup analysis to look at the various dosage levels studied for all three primary outcomes.

The *a priori* sensitivity analyses for the primary outcomes were: (1) *Best case -* Participants not completing follow-up or with inadequate seizure data were assumed to be responders in the lacosamide group and non-responders in the control group. For the primary safety outcome, participants not completing follow-up or with inadequate data were assumed to have continued in the trial in the lacosamide arms and discontinued

if in the control arm. (2) *Worst case* - Participants not completing follow-up or with inadequate seizure data were assumed to be non-responders in the lacosamide group and responders in the control group. For the primary safety outcome, participants not completing follow-up or with inadequate data were assumed to have discontinued due to adverse events in their respective lacosamide groups and to have stayed in if in the control group.

## RESULTS

### Study selection

A total of 11 reports involving 3 studies were identified for inclusion in the review. The search of Medline, EMBASE, and CENTRAL provided a total of 200 citations. The search for unpublished literature (expert survey, manufacturer request, clinical trial registries, and conference abstract proceedings) provided a total of 207 citations. After removing duplicates, 357 citations independently underwent abstract review and 77 citations were considered potentially relevant studies. Of the 77 full-text articles screened, 66 citations were excluded. Three randomized controlled trials (*Ben-Menachem et al., 2007*; *Chung et al., 2010b*; *Halász et al., 2009*) (located as 11 publications (*Massie, 2007*; *Kalviainen et al., 2007*; *Halász et al., 2006*; *Chung et al., 2007a*; *Chung et al., 2009b*; *Jatuzis et al., 2005*; *Ben-Menachem et al., 2005*; *Chung et al., 2007b*), that studied 1311 participants, met the inclusion criteria for this review. The weighted kappa statistic for inter-rater agreement on including or excluding potential trials was "excellent" [$k = 0.90$, 95% CI (0.83, 0.97)] (*Fleiss & Cohen, 1973*). See flow diagram Fig. 1.

### Study characteristics

See Table 1 for the characteristics of the included studies and Table S2 for the table of excluded studies.

#### *Methods*

All three studies selected for the review were randomized, controlled, parallel group studies published in English. The duration of the intervention was 18 weeks for the *Ben-Menachem et al. (2007)* and *Chung et al. (2010b)* trials and 16 weeks for the *Halász et al. (2009)* trial. All trials had an 8-week monitoring period before baseline and a 2-week taper or transition to off or open-label continuation of lacosamide at the end of the maintenance phases. The maintenance phase extension trials (*Husain et al., 2011*; *Rosenfeld et al., 2011b*; *Rosenow et al., 2011*) did not meet the criteria for inclusion in this review and are not considered further.

#### *Participants*

The included studies involved 1311 randomized participants from Australia, Europe, and the USA. Three participants in the Ben-Menachem trial (*Ben-Menachem et al., 2007*) were removed from the study after randomization for protocol violations and it could not be determined which dosage arm they belonged to. Patients were included in

**Table 1** Characteristics of included studies.

| First author & Publication year | Methodologic Quality | N | Patients — Characteristics | Intervention & comparator | Outcomes — Primary | Outcomes — Secondary | Funding |
|---|---|---|---|---|---|---|---|
| Ben-Menachem et al. (2007) | Adequate sequence generation[*]; AC; Blinding of patients, physicians, outcome assessors and data collectors; not ITT; incomplete reporting of pre-specified outcomes; follow-up to 18 weeks | 421 | **Mean age (SD)** 39.9 (11.3) **Gender:** 54% female **Concomitant AEDs:** 84% of the population were taking 2 AEDs at baseline, the rest were on 1 AED **Median seizure frequency per 28 days across all treatment groups during the baseline period:** 12 | • Lacosamide 100 mg PO BID • Lacosamide 200 mg PO BID • Lacosamide 300 mg PO BID • Placebo PO BID **Duration of treatment:** 18 weeks (after 8 week baseline monitoring - 6 week dose-titration & 12 week | • Change in seizure frequency per 28 days from baseline to maintenance • 50 % responder rate **Outcomes assessed at:** Weeks 0 & 18 | • Adverse event (AE) data: including serious adverse events, and discontinuation due to AEs • Achievement of seizure-free status Efficacy outcomes assessed at: weeks 0 & 18 • QOL scales (CGIC & QOLIE-31 – only in UK & USA) assessed at Week 0, 6, & 18 • Adverse effects (assessed Weekly 0–6 weeks and 10, 14 and 18 weeks) | Schwarz Bio-sciences Inc. |
| Chung et al. (2010b) | Adequate sequence generation; AC; Blinding of patients, physicians, outcome assessors and data collectors; not ITT; incomplete reporting of pre-specified outcomes follow-up to 18 weeks | 405 | **Mean Age (SD):** 38.3 (12.1) **Gender:** 50.6% female **Concomitant AEDs:** Throughout the trial 82.1% were taking 2–3 concomitant AEDs **Median seizure frequency per 28 days across all treatment groups during the baseline period:** P 15.0 L400 11.5 L600 16.5 | • Lacosamide 200 mg PO BID • Lacosamide 300 mg PO BID • Placebo PO BID **Duration of treatment:** 18 weeks (after 8 week baseline monitoring - 6 week titration & 12 week maintenance phase) | • Change in seizure frequency per 28 days from baseline to maintenance • 50 % responder rate **Outcomes assessed at:** Week 0 & 18 | • Adverse event (AE) data: including serious adverse events, and discontinuation due to AEs • % change in seizure frequency per 28 days from baseline to maintenance • 75% responder rate (the proportion of patients who experienced a 75% or greater reduction in seizure frequency from baseline to maintenance • Number & proportion of patients achieving seizure-free status throughout the maintenance period for patients completing the maintenance period and having complete efficacy data • Change in seizure frequency and 50% responder rate differentiated by seizure type **Adverse effects** **Outcomes assessed at:** weeks 0 & 18 | Schwarz Bio-sciences Inc, UCB Group |

*(continued on next page)*

Table 1 (*continued*)

| First author & Publication year | Methodologic Quality | Patients | | Intervention & comparator | Outcomes | | Funding |
|---|---|---|---|---|---|---|---|
| | | N | Characteristics | | Primary | Secondary | |
| *Halász et al. (2009)* | Adequate sequence generation; AC; Blinding of patients, physicians, outcome assessors and data collectors; incomplete ITT, not ITT, incomplete reporting of pre-specified outcomes, follow-up to 16 weeks | 485 | **Mean Age (SD):** 37.8 (11.9) **Gender:** 48.5% female **Concomitant AEDs:** 37% were taking 3 AEDs, 50% were taking 2 AEDs and 13% were taking 1 AED in addition to the trial medication **Median seizure frequency per 28 days across all treatment groups during the baseline period:** P 9.9 L200 11.5 L400 10.3 | • Lacosamide 100 mg PO BID • Lacosamide 200 mg PO BID • Placebo PO BID **Duration of treatment:** 16 weeks (after 8 week baseline - 4 week titration and 12 week maintenance phase) | • Change in seizure frequency per 28 days from baseline to maintenance • 50 % responder rate **Outcomes assessed at:** weeks 0 & 16 | • Number & Proportion of patients achieving seizure-free status through the maintenance period for patients completing the maintenance period • Proportion of seizure-free days during the maintenance period for patients entering the maintenance period **Efficacy Outcomes assessed at:** weeks 0 & 16 • **Adverse effects** **Outcomes assessed:** weekly 0–16 weeks • QOL scores (PGIC, CGIC, SSS, QOLIE-31) **QOL Outcomes assessed at:** weeks 0 & 18 | UCB Group |

**Notes.**

AC = allocation concealed, ITT = intention-to-treat analysis; N = total number of patients randomized; P = placebo; PO = oral; BID = twice daily; L200 = lacosamide 200 mg/day; L400 = lacosamide 400 mg/day; L600 = lacosamide 600 mg/day; CGIC = Clinical Global Impression of Change score; QOL – quality of life; QOLIE-31 = quality of life in epilepsy; PGIC = Patient's Global Impression of Change Score; SSS = seizure severity scale.

\* Randomization method or details not provided by author/manufacturer.

these studies if they had a diagnosis of partial-onset seizures (with or without secondary generalizations) that was objectively confirmed (with electroencephalogram (EEG) and magnetic resonance imaging (MR) or computed tomography (CT) scan). In order to be eligible, patients must have had partial-onset seizures for at least the previous two years despite treatment with at least two AEDs. For all three trials, to be counted as having "current seizures", participants must have had at least 4 partial-onset seizures per 28 days on average with no seizure-free period longer than 21 days. For the Ben-Menachem trial, the above inclusion criteria applied to the 8 week baseline period, whereas in the *Chung et al. (2010b)* and *Halász et al. (2009)* trials, the seizure frequency criteria also applied to the 8 weeks prior to baseline. All patients needed to have stable AED regimens for the 4 weeks prior to enrollment and the baseline period. In the Ben-Menachem trial, regimens could be 1 or 2 AEDs with or without vagal nerve stimulation (VNS). In the Chung et al and Halász et al trials, patients' regimens could consist of 1–3 AEDs with or without VNS. Participant age was restricted to over 16 years in two trials (*Chung et al., 2007b*; *Halász et al., 2009*) and over 18 years in one trial (*Ben-Menachem et al., 2007*). Pediatric data was not presented separately in the two studies that included patients less than 18 years of age.

### *Intervention*

All three studies compared adjunctive oral lacosamide in multiple doses to placebo (no active comparators) in a minimum of three comparator arms. All three trials had a lacosamide 200 mg twice daily arm. Ben-Menachem et al. and Halász et al. both had lacosamide 100 mg twice daily arms. Chung et al and Ben-Menachem included a lacosamide 300 mg twice daily arm. No studies included intravenous lacosamide.

### *Outcomes*

The primary outcomes for the three studies were change in seizure frequency (per 28 days from baseline to the maintenance period) and 50% responder rate. All three publications reported 50% responder rate in percentage, so the efficacy analysis denominators were used to convert to the number of patients who achieved the 50% response rate. Discontinuation due to adverse events was reported in all studies, as were individual adverse events. If percentages were provided for adverse event endpoints, they were converted to numbers of patients experiencing an event using the denominators provide for the safety analysis in the full publications. Quality of life outcomes were measured by two of the three studies (*Ben-Menachem et al., 2007*; *Halász et al., 2009*) , but only reported by *Ben-Menachem et al. (2007)*. Timing of outcome measures varied with the end of the maintenance period as defined by the individual studies.

## Risk of bias within studies

See Fig. 2.

All three studies were randomized-controlled trials, and all studies except Ben-Menachem presented the method of random sequence generation. Allocation concealment and blinding of participants, personnel, and outcome assessors were adequately reported for all trials. Incomplete outcome data reporting was present for all three trials. Selective outcome reporting was noted for all three included trials, as assessed by comparison of

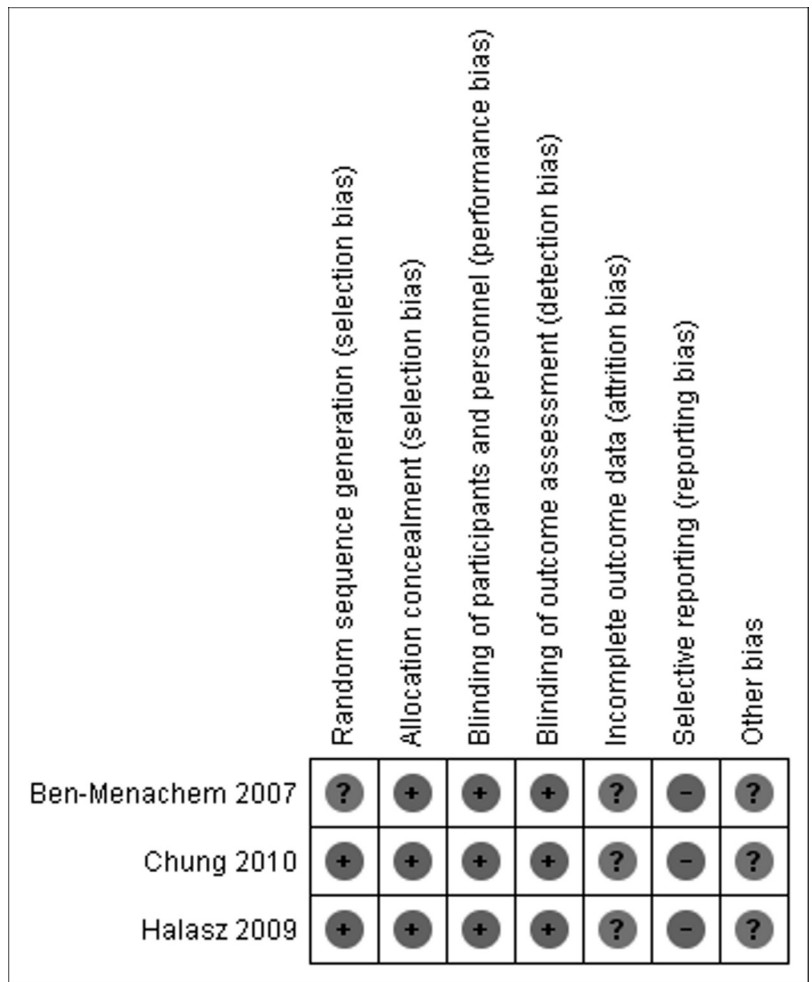

**Figure 2 Risk of bias summary.**

the published trials to the studies submitted for FDA approval. None of the three trials did a formal ITT analysis, but used all patients who received at least one dose of study medication as their definition of the study population.

## Results of individual studies

### Primary outcomes

The mean change in seizure frequency from maintenance phase to baseline was not provided in any of the three included studies. The authors of each study were contacted in an attempt to procure the seizure frequency change data, but no information was provided.

The primary outcome data available from the three trials ("percent reduction in seizure frequency") is presented in Table 2. There was a larger median percent change (as noted by the trial authors) with the higher dosage arms of lacosamide compared to placebo.

The 50% responder rate was reported for all three included trials and the results are presented in Fig. 3. In the meta-analysis of this primary outcome (ITT), lacosamide

**Table 2  Median percentage reduction in seizure frequency.**[*]

| Trial | Placebo | | Lacosamide 200 mg/day | | Lacosamide 400 mg/day | | Lacosamide 600 mg/day | |
|---|---|---|---|---|---|---|---|---|
| | N | % | N | % | N | % | N | % |
| Ben-Menachem et al. (2007) | 96 | 10% | 107 | 26% | 107 | 39% | 105 | 40% |
| Chung et al. (2010b) | 104 | 20.8% | – | – | 201 | 37.3% | 97 | 37.8% |
| Halász et al. (2009) | 159 | 20.5% | 160 | 35.3% | 158 | 36.4% | – | – |

**Notes.**

[*] Compares maintenance phase to baseline period.

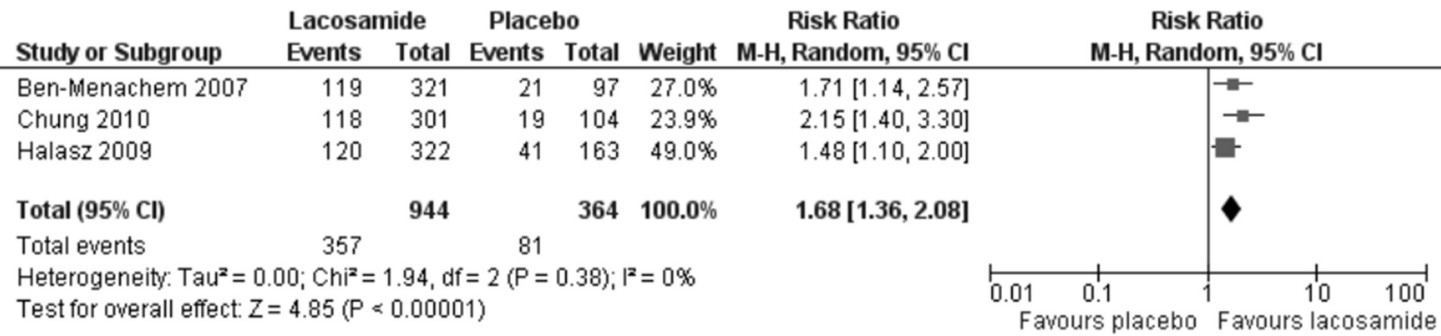

**Figure 3  50% responder rate (ITT).** Primary outcome (ITT) lacosamide (all dosage arms pooled) compared to placebo.

(all dosage arms pooled) was associated with a significantly higher 50% response rate compared to placebo (RR 1.68, 95% CI 1.36, 2.08). There was no evidence of statistical heterogeneity ($I^2 = 0\%$). The analyses of worst-case scenarios and best –case scenarios both produced similar results to the base analysis (RR = 1.62, 95% CI 1.24, 2.11; $I^2 = 37\%$) and (RR = 1.73, 95% CI 1.40, 2.13; $I^2 = 0$), respectively.

Discontinuation of study drug due to adverse events was reported in all three trials. In the meta-analysis of this outcome (ITT), lacosamide (all dosage arms pooled) was associated with a significantly higher rate of discontinuation with an RR 3.13 (95% CI 1.94, 5.06). There was no evidence of heterogeneity ($I^2 = 0\%$), see Fig. 4. Best-case and worst-case scenarios were not calculated for this outcome as no patient data was missing.

## Secondary outcomes

### Adverse effects outcomes

Statistically significant changes (higher rates in the lacosamide pooled dosage arm) were seen in the following adverse event outcomes: ataxia (RR 5.03, 95% CI 2.23, 11.37, see Fig. 5), dizziness (RR 3.49, 95% CI 2.43, 5.01, see Fig. 6), fatigue (RR 2.04, 95% CI 1.08, 3.85, see Fig. 7) and nausea (RR 2.36, 95% CI 1.22, 4.58, see Fig. 8). No heterogeneity was found in any of the adverse events ($I^2 = 0$), except for nausea, which showed moderate inconsistency, with an $I^2 = 34\%$. For the outcome of ataxia, data were included if outcomes were reported as ataxia or "coordination abnormal".

**Figure 4** **Discontinuation due to adverse events (ITT).** Lacosamide (all dosage arms pooled) compared to placebo.

**Figure 5** **Ataxia.** Lacosamide (all dosage arms pooled) compared to placebo.

**Figure 6** **Dizziness.** Lacosamide (all dosage arms pooled) compared to placebo.

All other meta-analyses and forest plots including: other adverse event outcomes (headache, somnolence, serious adverse events), seizure-free during the treatment period, the sensitivity analyses for best and worst case scenarios and the dose-response analyses can be found in Figs. S1–S10.

## Quality of life outcomes

The quality of life outcomes were incompletely reported across all included studies. Mean change in QOLIE-31 was reported in the Ben-Menachem trial (*Ben-Menachem et al., 2007*) but no measure of variance (SD) was provided. The measurement of QOLIE-31 was limited by language availability. Since the measurement scale was only available in

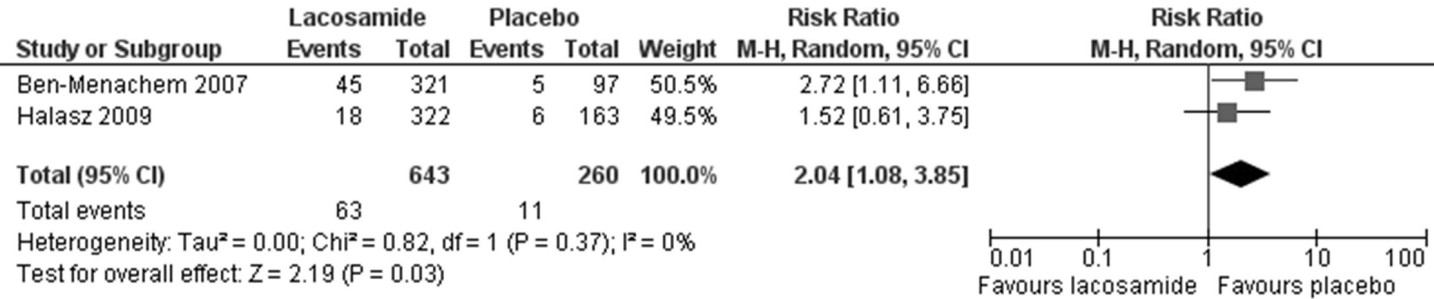

**Figure 7** **Fatigue.** Lacosamide (all dosage arms pooled) compared to placebo.

**Figure 8** **Nausea.** Lacosamide (all dosage arms pooled) compared to placebo.

**Table 3** **Quality of life outcomes.** Mean change in QOLIE-31 as reported in the *Ben-Menachem et al. (2007)* trial.

|  | Placebo | Lacosamide 200 mg | Lacosamide 400 mg | Lacosamide 600 mg |
|---|---|---|---|---|
| QOLIE-31 median change in overall score from baseline | −1.3 points | NR | 2.7 points | NR |
| Clinical Global Impression of Change (CGIC) | 25% | 35% | 40% | 38% |

**Notes.**
An increase in score indicates an improvement in quality of life as measured by the score. QOLIE-31 = quality of life in epilepsy inventory.

English, only participants from the United Kingdom or the United States of America were able to contribute to this outcome. The Clinical Global Impression of Change (CGIC) score was not reported as a continuous outcome (mean change), but as a dichotomous outcome from "Very much improved" or "much improved" from baseline to maintenance. No denominators for the groups were provided. There was a numerically larger change in seizure frequency in the lacosamide arms compared to the placebo arms. Although *Halász et al. (2009)* reported that they would measure quality of life outcomes; these were not reported in the final publication. The quality of outcome scales as reported by *Ben-Menachem et al. (2007)* are provided in Table 3.

## Economic outcomes

No trials reported on hospital admission, length of stay, length of stay in a specialized epilepsy unit or economic outcomes.

## Additional analysis

The planned sensitivity analyses for the primary efficacy outcome and 50% responder rate were ultimately not undertaken due to the small number of studies and lack of information relating to the pre-specified subgroups.

Post-hoc exploratory analyses were undertaken to explore a dose-effect relationship with efficacy and safety for two of the primary outcomes. The different dosage arms of lacosamide vs. placebo were placed into separate subgroups to explore the dose response of the outcomes for 50% responder-rate and discontinuation of study drug. Both analyses showed that, as the lacosamide dose increased, so too did the trend in the 50% response rate (P for interaction = 0.26) (Fig. S7) and the discontinuation of study drug due to adverse events (P for interaction = 0.03) (Fig. S8).

As a post-hoc analysis, we also pooled the lacosamide 100 mg BID and 200 mg BID arms (200 mg or less) and compared them to the 300 mg BID arm (greater than 200 mg). These analyses (Figs. S9 and S10) support the previous finding that higher doses of lacosamide are associated with greater risk of drug discontinuation due to adverse events (RR 2.28, 95% CI 1.46, 3.58; $I^2 = 0\%$).

## Risk of bias across studies

No statistical heterogeneity ($I^2 = 0\%$) was found in the analyses of the primary outcomes and in the majority of the outcomes assessed in the meta-analysis. Funnel plot asymmetry was not tested because only three studies were included in this meta-analysis, rendering this test unreliable (*Sterne, Egger & Moher, 2011*).

Where heterogeneity did exist in the pre-specified analyses (nausea, 50% responder rate worst-case scenario analysis), it was small to moderate with non-significant p-values. Potential sources of heterogeneity could be: (1) the differing dosage arms of lacosamide used in the trials and, (2) the differing lengths of treatment (dose-titration was 4 weeks in one trial and 6 in the other two trials).

Selective reporting was evident in all three studies included in this review. Consulting the FDA approval documents provided a more complete list of outcomes to be measured in the trials but no details could be obtained on many outcomes, including one of the pre-defined primary outcomes - change in seizure frequency from baseline. This was the stated primary outcome in all three trials, but was not reported in any of the publications.

The quality of evidence was downgraded for risk of bias and imprecision where appropriate (*Guyatt et al., 2008*). See the Summary of Findings for the statistically significant outcomes from this review (Table 4).

**Table 4 Summary of findings.** Statistically significant outcomes from the review graded for risk of bias and imprecision using GRADE.

| Outcomes | Illustrative comparative risks* (95% CI) | | Relative effect (95% CI) | No of Participants (studies) | Quality of the evidence (GRADE) |
|---|---|---|---|---|---|
| | Assumed risk Placebo | Corresponding risk Lacosamide | | | |
| **50% Responder Rate (ITT) - Lacosamide (all) vs. placebo** Follow-up: 16–18 weeks | **223 per 1000** | **374 per 1000** (303 to 463) | **RR 1.68** (1.36 to 2.08) | 1308 (3 studies) | ⊕⊕⊕ **moderate**[abc] |
| **Discontinuation of Study Drug due to Adverse Effects (ITT) - Lacosamide (all) vs. placebo** Follow-up: 16–18 weeks | **49 per 1000** | **155 per 1000** (96 to 250) | **RR 3.13** (1.94 to 5.06) | 1308 (3 studies) | ⊕⊕⊕ **moderate**[abcd] |
| **Ataxia** Follow-up: 16–18 weeks | **16 per 1000** | **83 per 1000** (37 to 187) | **RR 5.03** (2.23 to 11.37) | 1308 (3 studies) | ⊕⊕⊕ **moderate**[abcde] |
| **Dizziness** Follow-up: 16–18 weeks | **80 per 1000** | **278 per 1000** (194 to 399) | **RR 3.49** (2.43 to 5.01) | 1308 (3 studies) | ⊕⊕⊕ **moderate**[abcd] |
| **Fatigue** Follow-up: 16–18 weeks | **42 per 1000** | **86 per 1000** (46 to 163) | **RR 2.04** (1.08 to 3.85) | 903 (2 studies) | ⊕⊕⊕ **moderate**[abcd] |
| **Nausea** Follow-up: 16–18 weeks | **44 per 1000** | **104 per 1000** (54 to 201) | **RR 2.36** (1.22 to 4.58) | 1308 (3 studies) | ⊕⊕⊕ **moderate**[abcd] |

**Notes.**

GRADE Working Group grades of evidence (*Guyatt et al., 2008*): **High quality:** Further research is very unlikely to change our confidence in the estimate of effect. **Moderate quality:** Further research is likely to have an important impact on our confidence in the estimate of effect and may change the estimate. **Low quality:** Further research is very likely to have an important impact on our confidence in the estimate of effect and is likely to change the estimate. **Very low quality:** We are very uncertain about the estimate.

* The basis for the **assumed risk** (e.g., the median control group risk across studies) is provided in footnotes. The **corresponding risk** (and its 95% confidence interval) is based on the assumed risk in the comparison group and the **relative effect** of the intervention (and its 95% CI). **CI:** Confidence interval; **RR:** Risk ratio.

[a] Trials all had selective reporting of outcomes – outcomes reported in the protocol documents found in the FDA files did not match the outcomes reported in the peer-review publication.

[b] All 3 trials stated "double-blind" without further explanation given. Blinding was assessed as per Akl et al. Journal of Clinical Epidemiology. 2012; 65: 262–267.

[c] None of the 3 trials adhered to the intention-to-treat (ITT) principle - but performed "ITT" analysis with all patients who received at least one dose of study medication.

[d] Ben-Menachem trial provided no explanation with regards to their random sequence generation.

[e] Total number of events less than 300, based on Mueller et al. Annals of Internal Medicine 2007; 146: 878–881.

[f] Wide confidence intervals suggest a degree of imprecision.

## DISCUSSION

### Summary of evidence

Overall, the evidence from three included trials supports that lacosamide improves the 50% responder rate compared to placebo in adults with partial-onset seizures. The reduction in seizures demonstrated by this efficacy outcome must be weighed against the increased rate of discontinuation due to adverse events and the risk of increased ataxia, dizziness, fatigue, and nausea.

Post-hoc analysis demonstrated a possible dose-response relationship with 50% responder rate. While post-hoc analyses should always be interpreted cautiously, the test for interaction for discontinuation of study drug due to adverse events was statistically significant ($p = 0.03$), indicating that this is an important area for future research.

Incomplete outcome reporting impacted the final results of this systematic review. As study authors did not report the mean seizure frequency in each group, no quantitative analyses could be conducted on the change in seizure frequency, one of the *a priori* primary outcomes of this review. Quality of life outcomes were only selectively reported, and the data available for this review were not amenable to meta-analysis. All of the authors mentioned that lacosamide had a favourable effect on QOL measures.

None of the included trials reported on economic outcomes. From a strict drug cost perspective, lacosamide is far more expensive compared to other available AEDs (Table S3). Comparative trials with other AEDs would be helpful in determining the most cost effective role in the treatment of seizures.

The findings from this review can be directly applied to ambulatory patients with partial epilepsy, who are refractory to their current AED therapy. Hospitalized patients were not included in any of the trials in this review.

### Limitations

This systematic review used a robust search strategy to consider all of the best available published and unpublished evidence of lacosamide in partial-onset seizures in adults; however there were a few limitations to this review. The post-hoc analyses indicate the possible presence of a dose-response for efficacy and adverse effects for lacosamide. The pooling of all lacosamide doses as the comparator arm compared to placebo could underestimate the benefit of the higher doses and/or overestimate the effect of the lower doses with respect to efficacy. This pooling of all dose arms of lacosamide would be expected to similarly affect the adverse events outcomes if a dose-response relationship exists.

All of the included studies (which formed the basis of regulatory approvals) were of a very short duration (three months of maintenance therapy), which may have exaggerated the efficacy of lacosamide as adjunctive AED therapy, given that these therapies are usually administered for many years. These regulatory approval trials tend to have limited external generalizability due to exclusion of patients with co-morbidities which are common in a large percentage of epileptic patients (*Chung et al., 2009b*; *Jatuzis et al., 2005*).

The search strategy was not designed to retrieve economic analyses, so relevant economic studies on lacosamide could have been missed.

Incomplete reporting of outcomes precluded meta-analysis of the mean change in seizure frequency, one of the primary outcomes of this review. Overall, included studies were of moderate quality, as selective outcome reporting, using non-intention-to-treat analyses, and imprecision resulted in a downgrade in the quality of evidence of these randomized controlled trials. In this review, the risk for reporting bias, as evidenced by selective outcome reporting for all included trials, is judged to be the biggest threat to validity. Since only three studies were included, publication bias could not be explored.

This review does not apply to the pediatric population, as the two trials including patients less than 18 years old (*Chung et al., 2010b*; *Halász et al., 2009*) did not provide separate data on the pediatric participants.

### Implications for future research

Future research (both randomized controlled trials and systematic reviews/meta-analyses) should consider the following:

(1) The pediatric population remains largely unstudied and should be addressed as a distinct subgroup of patients with special attention to adverse events.

(2) The relative efficacy and safety of lacosamide in comparison to alternative AEDs has not been prospectively studied and is critically important to best inform clinical decision-making.

(3) Three cost-effectiveness analyses (European health care payer perspective) were found (*Simoens et al., 2010*; *Soini, Martikainen & Vanoli, 2009*; *Bolin, Berggren & Forsgren, 2010*). These analyses do not have direct applicability to the Canadian health care system. Additional cost-effectiveness analyses using multiple perspectives (including Provincial health system payers and society) are required in order to support effective decision making within the context of the Canadian Health Care system.

(4) Antiepileptics (such as lacosamide), while efficacious, are not a cure for epilepsy and can have wide-ranging side effects for patients. To better understand the implications of lacosamide therapy in the life of a patient with epilepsy, quality of life assessments and results should be reported completely. The results of the post-hoc dose-response analyses in this review warrant further *a priori* exploration with respect to safety and efficacy both in future RCTs and systematic reviews.

## CONCLUSIONS

This review provides evidence that lacosamide as adjunctive therapy in adult patients with partial-onset seizures increases the 50% responder rate, but with significantly more adverse events compared to placebo. The results are in agreement with the previously-published pooled studies and meta-analyses (*Beydoun et al., 2009*; *Chung et al., 2010a*; *Costa et al., 2011*; *Ryvlin, Cucherat & Rheims, 2011*) whilst providing a more accurate (ITT) summary estimate of benefit for lacosamide and a detailed look at risk of individual adverse events.

## ACKNOWLEDGEMENTS

The authors would like to thank Kaitryn Campbell for her valuable feedback and suggestions in constructing the search strategy and Dr. Elie Akl for his valuable methodological advice throughout this review.

### Funding

The evidence synthesis upon which this article was based was not funded. Support for the authors was provided by their employers (Pharmacy Department and Department of Anesthesia & Perioperative Medicine, London Health Sciences Centre, London, Ontario, Canada). The funders had no role in study design, data collection and analysis, decision to publish, or preparation of the manuscript.

### Grant Disclosures

The following grant information was disclosed by the authors:
Pharmacy Department and Department of Anesthesia & Perioperative Medicine, London Health Sciences Centre, London, Ontario, Canada.

### Competing Interests

Sonja Sawh, Jennifer Newman and Santosh Deshpande have no competing interests to declare. Dr. Philip M. Jones is an Academic Editor for PeerJ and has received research funds to participate in industry-sponsored studies by Merck and Hospira. These funds were used to pay study assistant salaries, with any remainder strictly used for investigator-initiated research. No personal remuneration has been received from any source.

### Author Contributions

- Sonja C. Sawh conceived and designed the experiments, analyzed the data, wrote the paper.
- Jennifer J. Newman provided assistance with the title and abstract screening, full-text screening, data extraction, editing assistance, methods and analysis discussion participation when needed.
- Santosh Deshpande and Philip M. Jones provided editing assistance, methods and analysis discussion participation when needed.

### Supplemental Information

Supplemental information for this article can be found online at http://dx.doi.org/10.7717/peerj.114.

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
