# Peer review of "Lacosamide adjunctive therapy for partial-onset seizures: a meta-analysis"

_PeerJ, doi:10.7717/peerj.114_

## Round 0.1 · original submission · Minor Revisions

· Academic Editor

Minor Revisions

The suggested revisions are very limited and largely confined to modification of the discussion text which I expect will present little difficulty. I look forward to receiving the revised MS in due course.

·

Basic reporting

Sections and references do not follow the correct style.

Experimental design

There are no specific comments on experimental design. Methods are adequate for the objective.

Validity of the findings

The validity of the findings is limited as clearly indicated in the MS. However, they have a good relevance despite other studies will be needed to draw proper conclusions on drug efficacy, safety and from an economical point of view.

Additional comments

Comments for 2013:05:531:0:1:REVIEW
Lacosamide adjunctive therapy for partial-onset seizures: a meta-analysis

Overall, I found the MS very interesting, only few minor points need to be addressed.
1. RCT in the abstract should not be abbreviated.
2. The declared objective of the study was to compare lacosamide efficacy and safety to other AEDs. However, authors results do not allow comparison if not only to placebo. Please modify the objective both in the abstract and introduction
3. Authors mention the fact that in two studies, patients between 16 and 18 years of age were included. It is not possible to identify them and therefore they were considered for the present analysis. Would this have any impact on authors’ results? This point should be addressed in the discussion limitations heading.

·

Basic reporting

Sawh and colleagues present a meta-analysis of the randomised controlled trial of lacosamide. They conclude that lacosamide significantly increases the probably of improvement (>50% reduction) in comparison at the expense of a significantly greater risk of side effects. I have only a couple of minor points to raise.

Experimental design

The methodology is adequate, but given the small number of studies included, this section could be shortened possibly using supplementary material.

Validity of the findings

The authors do not present the results in term of seizure freedom; they would also provide an important insight in the relevance of lacosamide in clinical practice.

In the discussion, the authors correctly acknowledged that regulatory trials are limited by their short duration. They should also discuss that regulatory trials results are also difficult to extrapolate to clinical practice as they tend to enrol people with a high seizure frequency to demonstrate efficacy quickly, and tend to exclude young people without significant co-morbidities (such intellectual disabilities or psychiatric conditions, not uncommon in epilepsy). Those studies assess therefore a specific population and are unlikely to represent the whole population of people with epilepsy candidate to this treatment.

---

## Round 0.2 · accepted · Accept

· Academic Editor

Accept

My thanks for undertaking the reviewers' suggested changes. With regard to the rebuttals made regarding methodology, I concur with you.